# Does Risk-Adapted Proton Beam Therapy Have a Role as a Complementary or Alternative Therapeutic Option for Hepatocellular Carcinoma?

**DOI:** 10.3390/cancers11020230

**Published:** 2019-02-15

**Authors:** Tae Hyun Kim, Joong-Won Park, Bo Hyun Kim, Hyunjung Kim, Sung Ho Moon, Sang Soo Kim, Sang Myung Woo, Young-Hwan Koh, Woo Jin Lee, Dae Yong Kim, Chang-Min Kim

**Affiliations:** 1Center for Liver Cancer, Research Institute and Hospital, National Cancer Center, Goyang 410-769, Korea; jwpark1@ncc.re.kr (J.-W.P.); bohkim@ncc.re.kr (B.H.K.); wsm@ncc.re.kr (S.M.W.); mrikyh@ncc.re.kr (Y.-H.K.); lwj@ncc.re.kr (W.J.L.); cmkim53@ncc.re.kr (C.-M.K.); 2Center for Proton Therapy, Research Institute and Hospital, National Cancer Center, Goyang 410-769, Korea; 12777@ncc.re.kr (H.K.); shmoon@ncc.re.kr (S.H.M.); sangsookim@ncc.re.kr (S.S.K.); radiopia@ncc.re.kr (D.Y.K.)

**Keywords:** hepatocellular carcinoma, proton beam therapy, overall survival

## Abstract

To evaluate the role of risk-adapted proton beam therapy (PBT) in hepatocellular carcinoma (HCC) patients, a total of 243 HCC patients receiving risk-adapted PBT with three dose-fractionation regimens (regimen A [*n* = 40], B [*n* = 60], and C [*n* = 143]) according to the proximity of their gastrointestinal organs (<1 cm, 1–1.9 cm, and ≥2 cm, respectively) were reviewed: The prescribed doses to planning target volume 1 (PTV1) were 50 gray equivalents (GyE) (EQD2 [equivalent dose in 2 Gy fractions], 62.5 GyE_10_), 60 GyE (EQD2, 80 GyE_10_), and 66 GyE (EQD2, 91.3 GyE_10_) in 10 fractions, respectively, and those of PTV2 were 30 GyE (EQD2, 32.5 GyE_10_) in 10 fractions. In all patients, the five-year local recurrence-free survival (LRFS) and overall survival (OS) rates were 87.5% and 48.1%, respectively, with grade ≥3 toxicity of 0.4%. In regimens A, B, and C, the five-year LRFS and OS rates were 54.6%, 94.7%, and 92.4% (*p* < 0.001), and 16.7%, 39.2%, and 67.9% (*p* < 0.001), respectively. The five-year OS rates of the patients with the Modified Union for International Cancer Control (mUICC) stages I, II, III, and IVA and Barcelona Clinic Liver Cancer (BCLC) stages A, B, and C were 69.2%, 65.4%, 43.8%, and 26.6% (*p* < 0.001), respectively, and 65.1%, 40%, and 32.2% (*p* < 0.001), respectively. PBT could achieve promising long-term tumor control and have a potential role as a complementary or alternative therapeutic option across all stages of HCC.

## 1. Introduction

Various treatment options for hepatocellular carcinoma (HCC) are currently available, including liver transplantation, surgical resection, local ablative treatments such as radiofrequency ablation (RFA) and percutaneous ethanol injection (PEI), transarterial chemoembolization (TACE), radioembolization, and molecular targeted agents, such as sorafenib [1,2,3,4]. With technological advances in radiotherapy (RT), new modern RT techniques including three-dimensional conformal RT (CRT), intensity-modulated RT, stereotactic body RT (SBRT), and RT with proton beams and carbon ion beams have recently been applied for HCC patients with or without tumor vascular thrombosis (TVT) and have shown promising outcomes [5,6,7,8,9,10,11,12,13,14,15,16,17,18,19,20,21,22]. Proton beams, unlike X-rays, have unique physical properties, called Bragg peaks, which allow the delivery of a high dose of radiation to the target volume without exit dose to outside of the target volume. Our previous study showed that proton beam therapy (PBT) could spare the normal liver more effectively than RT with X-rays [23]. In addition, a recent meta-analysis [24] showed that charged particle therapy has higher rates of survival than CRT, similar to SBRT, and PBT tends to result in a lower incidence of adverse events than CRT and SBRT.

HCC patients have poor functional reserves resulting from underlying liver cirrhosis (LC), and primary tumors and/or TVT are often located near radiosensitive tissues, such as the gastrointestinal (GI) organs; thus, when RT is performed in HCC patients with or without TVT, it is important to spare both the remaining normal liver and GI organs. The PBT using simultaneous integrated boost (SIB) technique, which simultaneously delivers different doses to different targets, can potentially reduce irradiated doses to surrounding normal tissues and overall time of treatment and improve the therapeutic ratio compared to conventional fractionated PBT. Based on this rationale, risk-adapted PBT using the SIB technique has been used for HCC patients with or without TVT at our institution from June 2012. The purpose of this study was to evaluate the long-term efficacy and safety of risk-adapted PBT in these patients.

## 2. Results

A total of 314 patients treated with PBT for HCC from June 2012 to April 2017 were registered. Of these, 71 patients did not meet the eligibility criteria for the following reasons: 48 were treated with PBT using different dose-fractionation regimens, 11 had extrahepatic disease, four had uncontrolled intrahepatic disease outside of the PBT site, four had Child-Pugh class B8-9 or C, and four were transferred to other hospitals immediately after PBT. The remaining 243 patients who met all of the inclusion criteria were analyzed in this study; this included 122 patients who participated in the phase II trial for risk-adapted PBT using three dose-fractionation regimens (NCC20120622), and 121 patients who did not participate (Table 1). Because of the proximity of the GI structures, the frequencies of large tumor sizes, TVT, advanced tumor stage, concurrent sorafenib, and post-treatment to the PBT site were significantly lower in the order of regimen A, B, and C (*p* < 0.05) (Table 1).

Primary tumor and TVT responses for all patients were complete response (CR) in 199 (81.9%) and 30 (50.8%), respectively; partial response (PR) in 30 (12.3%) and 18 (30.5%), respectively; stable disease in 13 (5.3%) and 10 (16.9%), respectively; and progressive disease in 1 (0.4%) and 1 (1.7%), respectively (Table 2) (Figure 1A–I). Median times to CR of the primary tumor and TVT after PBT were 4.5 months (range 1–21.7 months) and 5.1 months (range 1.1–16.4 months), respectively. The CR and objective response (CR + PR) rates of primary tumor and TVT were significantly lower in regimen A than in regimens B and C (*p* < 0.05 each) (Table 2). Not surprisingly, the patients who had CR less frequently received post-treatment to the PBT site than those who did not have CR (25 of 199 [12.7%] vs. 16 of 37 [43.2%], *p* < 0.001) (Table 2).

At the time of analysis, 145 patients were alive, and 98 died by disease progression (*n* = 86), progressive liver failure induced by underlying LC (*n* = 8), pneumonia (*n* = 1), intracranial hemorrhage (*n* = 1), biliary complications by post-treatment (*n* = 1), and cardiomyopathy due to chemotherapy (*n* = 1) unrelated to PBT. The median follow-up time of all and living patients were 31.5 months (range 2.1–68.2 months) and 39.3 months (range 13.2–68.2 months), respectively. Disease recurrence occurred in 190 (78.2%) patients: 24 (12.6%) had local recurrence, 175 (92.1%) had intrahepatic recurrence, and 70 (36.8%) had distant metastases (Figure 2). In all patients, the median time of OS was 56.6 months (95% confidence interval [CI], 45.8–67.2 months) and the three- and five-year local recurrence-free survival (LRFS), recurrence-free survival (RFS), and overall survival (OS) rates were 88.6% (95% CI, 84.1–93.1%) and 87.5% (95% CI, 82.6–92.4%), and 18.8% (95% CI, 13.5–24.1%) and 12.4% (95% CI, 6.3–18.5%), and 61.8% (95% CI, 55.1–68.5%), and 48.1% (95% CI, 39.1–57.1%), respectively. In the patients treated with regimens A, B, and C, the five-year LRFS, RFS, and OS rates were 54.6% (95% CI, 32.6–76.7%), 94.7% (95% CI, 88.8–100%), and 92.4% (95% CI, 87.5–97.3%) (*p* < 0.001), 10% (95% CI, 0.2–19.8%), 12.9% (95% CI, 0.4–25.4%), and not reached (NR) (95% CI, -) (*p* = 0.038), and 16.7% (95% CI, 3.6–29.8%), 39.2% (95% CI, 24.9–53.5%), and 67.9% (95% CI, 57.5–78.3%), respectively (*p* < 0.001) (Figure 3A). In those who had CR or not, the five-year LRFS, RFS, and OS rates were 94% (95% CI, 89.7–98.35) and NR (*p* < 0.001), 14.9% (95% CI, 7.8–22%) and NR (95% CI, -) (*p* < 0.001), and 60.9% (95% CI, 51.5–70.3%) and 0% (*p* < 0.001), respectively (Figure 3B). In those with modified International Union Against Cancer (mUICC) stages I, II, III, and IVA, the five-year LRFS, RFS, and OS rates were 100% (95% CI, -), 94% (95% CI, 87.3–100%), 88.9% (95% CI, 82.2–95.6%), and 65.5% (95% CI, 58.8–88.6%) (*p* < 0.001), 23.4% (95% CI, 0–49.5%), 27.3% (95% CI, 15.5–39.1%), 0.4% (95% CI, 0–6.9%), and 11.8% (95% CI, 2.4–21.2%)(*p* < 0.001), and 69.2% (95% CI, 28.6–100%), 65.4% (95% CI, 45.8–85%), 43.8% (95% CI, 31.3–56.3%), and 26.6% (95% CI, 12.7–40.5%) (*p* < 0.001), respectively (Figure 3C). In those with Barcelona Clinic Liver Cancer (BCLC) stages A, B, and C, the five-year LRFS, RFS, and OS rates were 93.6% (95% CI, 87.9–99.3%), 89.7% (82.3–97.1%), and 72.3% (95% CI, 57–87.6%) (*p* = 0.001), 17.6% (95% CI, 7.8–27.4%), 0% (95% CI, -), and 16.4% (95% CI, 6.6–26.2%) (*p* = 0.022), and 65.1% (95% CI, 50.2–80%), 40% (95% CI, 26.1–53.9%), and 32.2% (95% CI, 16.0–48.6%) (*p* < 0.001), respectively (Figure 3D).

Except for gender, age, performance status, etiology of LC, diagnosis at PBT, and pre- and post-treatment to other sites, all clinical factors were significantly associated with OS in univariate analysis (Table 3). In multivariate analysis, the Child–Pugh classification, alpha-fetoprotein (AFP) level, mUICC stage, dose-fractionation regimens, and primary tumor response were significantly associated with OS (*p* < 0.05 each) (Table 3). 

Acute adverse effects within three months after PBT were easily manageable and did not cause discontinuation of the treatment course. Of the 243 patients, 10 (4.1%) and one (0.4%) experienced grade 1 and 2 elevated alanine aminotransferase without disease progression, respectively, and 213 (87.7%) had no change in their Child–Pugh scores, 19 (7.8%) had a 1-point decrease, 10 (4.1%) had a 1-point increase, and one (0.4%) had a 2-point increase due to biliary obstruction by tumor progression. Thirty-two (13.2%) and two (0.8%) patients experienced grades 1 and 2 leukopenia, respectively, and 19 (7.8%) and one (0.4%) experienced grades 1 and 2 thrombocytopenia, respectively. Of 40 patients treated with regimen A, late GI toxicities, defined as gastric or duodenal ulcers within the PBT field, were observed in five (12.5%) patients, including one (2.5%), three (7.5%), and one (2.5%) with grades 1, 2, and 3, respectively. None of the patients treated with regimens B and C experienced GI bleeding and ulcer, and treatment-related hepatic failure and death were not observed.

## 3. Discussion

Surgical resection, liver transplantation and local ablative treatments, such as RFA and PEI, have been considered as curative treatments [1,3,4]. In patients with BCLC stage A, these curative treatments may offer five-year OS rates of 50–70% [1,25,26,27]. However, these curative treatments have been performed in selected patients because of the multi-centricity of HCC development, patient comorbidities, the lack of transplant donors, bleeding tendency, unsuitable location, non-echogenicity, or large tumor sizes. Fukuda et al. [7] reported five-year LRFS and OS rates of 94% and 69%, respectively, in previously untreated BCLC 0/A HCC patients who received PBT with 66–77 GyE in 10–35 fractions. Similarly, the present study also showed that the risk-adapted PBT using 50–66 GyE in 10 fractions for BCLC A HCC patients who were unsuitable or ineffective with other loco-regional treatments (LRTs) resulted in five-year LRFS and OS rates of 93.6% and 65.1%, respectively, which was comparable to those in patients treated with curative treatments including surgical resection and local ablative treatments in data from our institutional cohort [27] and previous studies [1,25,26]. These results suggest that PBT can be considered an effective alternative option for BCLC A patients.

TACE is considered a first-line treatment for BCLC B HCC patients who have unresectable tumors and are unsuitable for local ablative treatments [1,3,4]. The expected median survival time is >30 months [1]. In our institutional cohort data [27], BCLC B patients had a median OS of 32.5 months and a five-year OS rate of 34.3%. The mUICC I/II and III patients who were treated with TACE had five-year OS rates of 55.2% and 25.2%, respectively. In the present study, risk-adapted PBT resulted in median OS of 37.1 months and five-year OS rates of 40% in BCLC B patients, and five-year OS rates of 69.2%, 65.4%, and 43.8% in mUICC I, II, and III patients. In BCLC C patients, although sorafenib has been considered a first-line treatment with an expected median OS of ≥10 months [1,25,26], TACE remains the most frequently used initial treatment [27,28,29]. In our institutional cohort data [27], BCLC C patients had a median OS of 10.3 months and five-year OS rates of 17.1%, and the mUICC IVA patients treated with TACE had a median OS of 7.8 months and five-year OS rates of 5.9%. The present study showed a median OS of 33.9 months and a five-year OS of 33.9% in BCLC C patients and a median OS of 19.4 months and a five-year OS of 26.6% in mUICC IVA patients. Although there were differences in patient characteristics and selection bias among the studies, these PBT results for BCLC B and C patients might be comparable or superior to those of any TACE and sorafenib. 

HCC lesion(s) and/or TVT are frequently adjacent to the GI organs. When considering RT for HCC patients, it is important to safely deliver a high dose of radiation to the tumor and/or TVT within tolerances of these organs. For these reasons, we applied risk-adapted PBT using three dose-fractionation regimens, depending on the proximity of the GI organs, and showed that risk-adapted PBT could be performed in all patients with grade ≥3 GI toxicity of 0.4%. Only one of 40 (2.5%) patients treated with regimen A (EQD2, 62.5 GyE_10_) for HCC <1 cm from the GI organs experienced grade 3 GI toxicity. No patients treated with regimens B (EQD2, 80 GyE_10_) and C (EQD2, 91.3 GyE_10_) for HCC 1–1.9 cm and ≥2 cm from the GI organs, respectively, experienced grade ≥3 GI toxicity. The five-year LRFS (54.6%, 94.7%, and 92.4%), RFS (10%, 12.9%, and NR), and OS rates (16.7%, 39.2%, and 67.9%) were significantly higher in the patients treated with regimens B and C (EQD2 ≥ 80 GyE_10_) than in those treated with regimen C (EQD2 < 80 GyE_10_) (Table 3). In other studies [20,22], PBT using relative long fractionation regimens (53.7–88 GyE_10_/24–38 fractions or 80.1–91.5 GyE_10_/22–35 fractions, respectively) showed a two-year LPFS rate of 94% and three-year LPFS of 88.1%, respectively, and grade ≥3 GI toxicity of 2.5% and 2.1%, respectively. In the present study, there were significant differences in patient characteristics including tumor size, stage, extent of TVT, etc., among three dose-fractionation regimens (Table 1). Thus, large HCC was more likely to close to GI organs and treat with regimen A rather than regimen B and C, and subsequently poor local tumor control in regimen A than regimen B and C. In addition, the incidence of grade ≥3 GI toxicity in the patients treated with regimen A was as low as 2.5%. These findings suggest that it might be possible to carefully escalate the radiation doses for patients with HCC <1 cm from the GI organs to improve tumor control and survival. 

This study had inherent limitations from single institutional retrospective data with a heterogeneous population including recurrent, ineffective, and/or unsuitable tumor with other LRTs. The effects of pre- and post-treatments on intrahepatic and/or metastatic disease and possible selection bias were not thoroughly assessed. Nevertheless, to date, the present study was a relatively large-size study treating patients with risk-adapted PBT using consistent techniques, and the outcomes of risk-adapted PBT were compared to those of current established therapeutic options, such as surgical resection, local ablative treatments, TACE, and sorafenib, with relatively large-scale institutional cohort data (*n* = 1972) [27]. In addition, tumor response and dose-fractionation regimens were significantly associated with LRFS and OS. These findings suggest that dose-escalation using risk-adapted PBT may improve LRFS and subsequently improve OS. 

## 4. Materials and Methods

### 4.1. Patients

Patients who were treated with PBT for primary or recurrent HCC from June 2012 to April 2017 were registered and the database was reviewed. Inclusion criteria for the present study were: (i) HCC was diagnosed by pathologic confirmation or radiologic findings plus serum alpha-fetoprotein (AFP) concentrations ≥200 ng/mL, according to the guidelines of the Korean Liver Cancer Study Group and the National Cancer Center (NCC) [3]; (ii) HCC lesions were unsuitable, ineffective, or refused for any other loco-regional treatments (LRTs), including surgical resection, RFA, TACE, PEI, etc.; (iii) patients were treated with risk-adapted PBT using three dose-fractionation regimens depending on the proximity of the GI organs; (iv) there was no uncontrolled intrahepatic disease outside of the targeted lesion; (v) liver function of Child–Pugh class A or B7 was present; (vi) no extrahepatic metastasis; and (vii) there was no previous history of RT to the liver. The Modified Union for International Cancer Control (mUICC) [30] and the Barcelona Clinic Liver Cancer (BCLC) [25] staging classification were used for tumor and clinical staging, respectively. This study was approved by the institutional review board of NCC (NCC20180100), and informed consent was not required because of the retrospective nature of this research. 

### 4.2. Treatment

The details of the risk-adapted PBT plan have been described in previous reports [11,13,14,15,17]. In brief, contrast-enhanced four-dimensional computed tomography (CT) scanning was performed under monitoring with a real-time position management (RPM) system (Varian Medical Systems, Palo Alto, CA, USA). All obtained CT images were resorted into 10 equally spaced respiratory phases and average intensity projection (AIP)-CT images were reconstructed with the exhalation (gated) phases (30% of total respiratory cycle) CT images. All tumors detected in the AIP-CT images were defined as the gross tumor volume (GTV) without clinical target volume margins from GTVs [5,11,13,14,15,17,31]. The sum of the GTVs, defined as the internal target volume (ITV), and the contours of the organs at risk (OARs) in each CT image during the gated phases were delineated. The planning target volumes (PTV) 1 and 2 included the ITV with a margin of 5–7 mm in all directions with and without excluding the 10-mm expanded volume of the GI organs, respectively (Figure 1D–F). Typically, PBT plans (Version 8.1; Varian Medical System, Palo Alto, CA, USA) were performed using one beam of 230 MeV proton beams of Proteus 235 (Ion Beam Applications, S.A., Louvain-la-Neuve, Belgium) to cover the PTV1 and two beams to cover the PTV2. The proximal, distal, border smoothing, smearing, and aperture margins of the proton beams with the double-scattering mode to PTV were set to 5–7 mm each. The radiation doses for the target volumes and the OARs were described in gray equivalents (GyE = proton physical dose [in gray] × relative biologic effectiveness [1.1]) and the equivalent dose in 2 Gy fractions (EQD2, GyE_10_, or GyE_3_) was calculated using a linear quadratic model with α/β of 10 and 3 for the acute and late effects, respectively (EQD2 = total dose × [(fraction dose + α/β)/(2 + α/β)]) [32]. The plan was designed so that at least 95% of the PTV would receive 100% of each prescribed dose. Three dose-fractionation regimens were used according to the proximity of the GI organs as described in our previous report [11]: (1) For regimen A, the prescribed doses to PTV1 and PTV2 were 50 GyE (EQD2, 62.5 GyE_10_) and 30 GyE (EQD2, 32.5 GyE_10_) in 10 fractions, 5 fractions a week, respectively, for the patients with GTV <1 cm from the GI organs; (2) for regimen B, the prescribed doses to PTV1 and PTV2 were 60 GyE (EQD2, 80 GyE_10_) and 30 GyE in 10 fractions, respectively, for the patients with GTV within 1–1.9 cm from the GI organs; and (3) for regimen C, the prescribed dose to PTV1, identical to PTV2, was 66 GyE (EQD2, 91.3 GyE_10_) in 10 fractions for the patients with GTV ≥2 cm from the GI organs. The dose-volume constraints for the OARS were described previously [11,13,14,15,17,33]. The relative volumes of the total and remaining normal liver receiving ≥27 GyE were below 60% and 50%, respectively, and the absolute volumes of the stomach and small and large bowel receiving ≥37 GyE and ≥35 GyE, respectively, were less than 2 cm^3^. At each treatment, to reduce intra- and inter-fractional uncertainties, all patients were asked to fast for at least 4 hours prior to treatment and irradiation was delivered during the gated phase after verifying each patient’s position and isocenter. 

### 4.3. Evaluation and Statistical Considerations

During PBT, the patients were evaluated weekly and, after completion of PBT, at 1 month, every 3 months for the first 2 years, and every 6 months thereafter. Physical examination, complete blood counts, liver-function tests, chest X-ray, and liver dynamic contrast-enhanced CT or magnetic resonance imaging were performed at each follow-up. TVT in the main, right, and left lobar branches of the portal vein, hepatic vein, and inferior vena cava were classified as “Main” and the other segmental and sectional branches of the portal vein and hepatic vein were classified as “Branch.” The responses of the primary tumor and TVT were defined as the maximal tumor response observed during the follow-up period comparing CT scans before and after PBT, according to the modified Response Evaluation Criteria in Solid Tumors criteria (mRECIST) [34]. Adverse effects related to PBT were graded according to Common Terminology Criteria for Adverse Events (CTCAE; version 4.0, https://evs.nci.nih.gov/ftp1/CTCAE/CTCAE_4.03/)

Recurrence was proven by pathologic and/or radiologic findings showing an increase in size over time. Local, intrahepatic, and distant recurrence was defined as a regrowth or new tumor within the treated volume, a regrowth or new intrahepatic tumor outside of the treated volume, and the development of an extrahepatic tumor, respectively. Overall survival (OS), recurrence-free survival (RFS), and local recurrence-free survival (LRFS) were estimated from the date of the start of PBT to the date of death or last follow-up, any recurrence, and local recurrence, respectively. The distributions of clinical characteristics among dose-fractionation regimens ware compared using Fisher’s exact test or one-way analysis of variance, and the differences of primary tumor and TVT responses according to dose-fractionation regimens were assessed using Fisher’s exact test. The probability of survival was calculated using the Kaplan–Meier method and, in univariate analysis, the log-rank test was used to evaluate the effects of the factors on survival. Multivariate analysis was performed using Cox’s proportional hazard model with a stepwise forward selection procedure. Differences with *p* values of <0.05 were considered statistically significant, and variables with *p* < 0.1 in univariate analysis were entered into multivariate analysis. All statistical analyses were performed using STATA software (version 14.0; StataCorp., College Station, TX, USA).

## 5. Conclusions

This study shows that risk-adapted PBT using three dose-fractionation regimens could achieve long-term tumor control with minimal toxicity, suggesting that PBT might play a potential role as a complementary or alternative therapeutic option across all stages of HCC that are unsuitable or ineffective with other LRTs. 

## Figures and Tables

**Figure 1 cancers-11-00230-f001:**
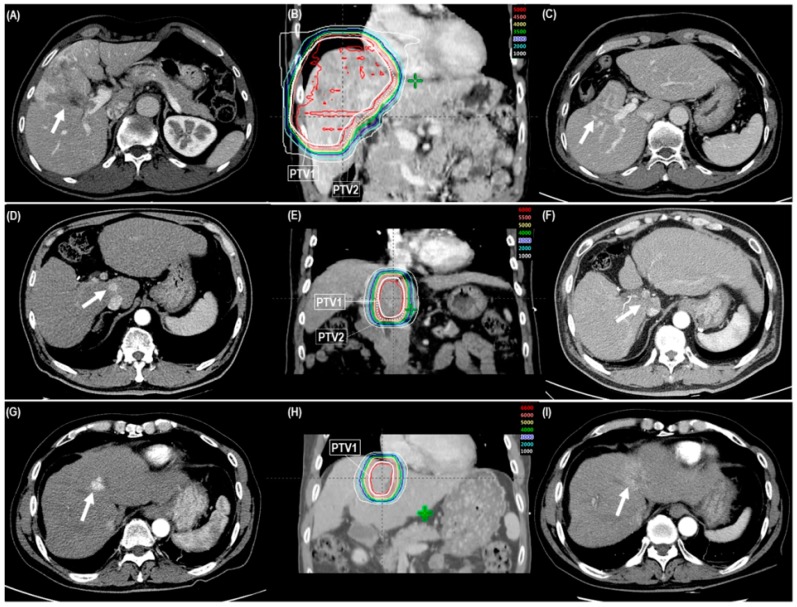
The cases showing objective tumor response after risk-adapted proton beam therapy (PBT) with three dose-fractionation regimens (regimen A, B, and C) according to the proximity of the gastrointestinal organs (<1 cm [**A**–**C**], 1–1.9 cm [**D**–**F**], and ≥2 cm [**G**–**I**], respectively). **A**, **D**, and **G**: Pre-treatment computed tomography (CT) scans showing the primary tumor and/or tumor vascular thrombosis (TVT) (arrow). **B**, **E**, and **H**: The patient received risk-adapted PBT (50 GyE, 60 GyE, and 66 GyE in 10 fractions to PTV1, respectively, and 30 GyE in 10 fractions to PTV2). **C**, **F**, and **I**: One-year follow-up CT scans after risk-adapted PBT showing notable reductions of the primary tumor and/or TVT (arrow). Abbreviations: PTV, planning target volume.

**Figure 2 cancers-11-00230-f002:**
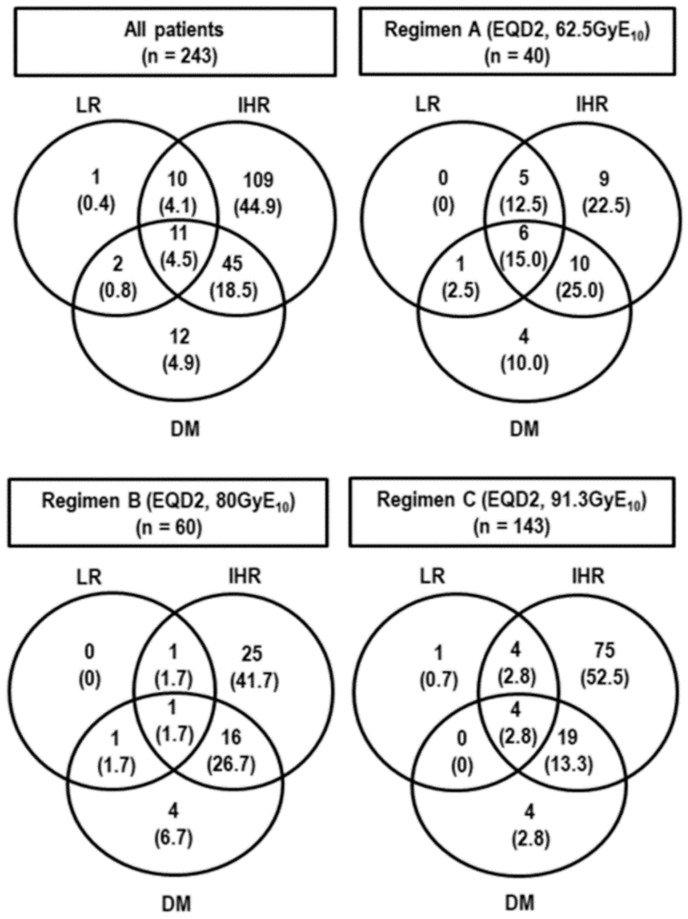
Patterns of failure. Abbreviations: LR, local recurrence; IHR, intrahepatic recurrence; DM, distant metastasis.

**Figure 3 cancers-11-00230-f003:**
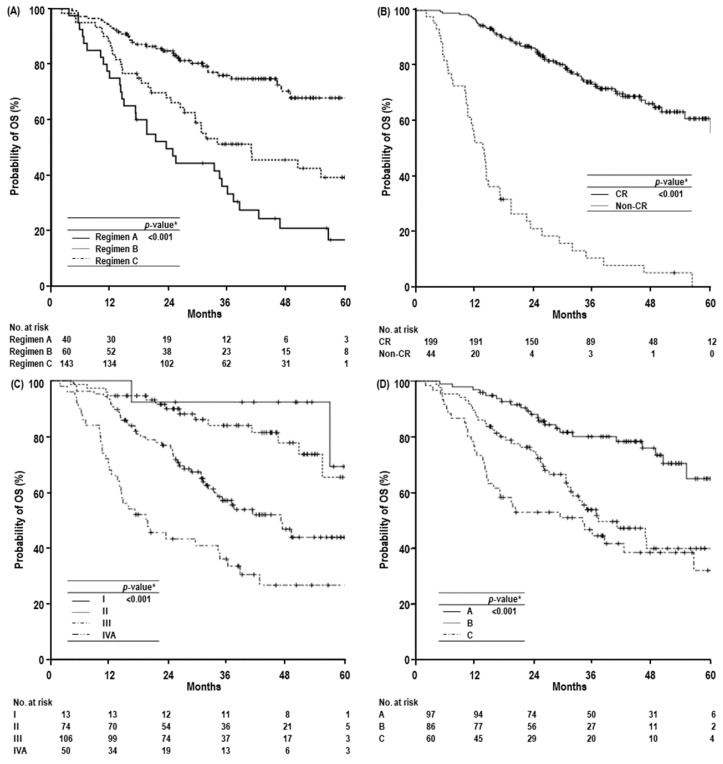
Overall survival (OS) curves according to dose-fractionation regimens (**A**), primary tumor response (**B**), mUICC stage (**C**), and BCLC stage (**D**). Abbreviations: mUICC stage, modified International Union against Cancer stage; BCLC stage, Barcelona Clinic Liver Cancer stage; yr, year; CI, confidence interval; CR, complete response. * Log-rank test.

**Table 1 cancers-11-00230-t001:** Patient characteristics in all patients and subgroups according to dose-fractionation (Dose-Fx) regimens.

Characteristics		Total	Dose-Fx Regimen A	Dose-Fx Regimen B	Dose-Fx Regimen C	*p* Value
		*n* (%)	*n* (%)	*n* (%)	*n* (%)	
Gender	Male	211 (86.8)	34 (85.0)	53 (88.3)	124 (86.7)	0.888 *
	Female	32 (13.2)	6 (15.0)	7 (11.7)	19 (13.3)	
Age, years	Median (range)	61 (24–92)	59 (24–81)	62.5 (39–80)	62 (34–92)	0.133 ^†^
	<60	100 (41.2)	22 (55.0)	24 (40.0)	54 (37.8)	0.144 *
	≥60	143 (58.8)	18 (45.0)	36 (60.0)	89 (62.2)	
ECOG PS	0	237 (97.5)	38 (95.0)	59 (98.3)	140 (97.9)	0.520 *
	1	6 (2.5)	2 (5.0)	1 (1.7)	3 (2.1)	
Etiology of LC	HBV	188 (77.4)	33 (82.5)	45 (75.0)	110 (76.9)	0.799 *
	HCV	20 (8.2)	3 (7.5)	6 (10.0)	11 (7.7)	
	Alcoholic	17 (7.0)	1 (2.5)	6 (10.0)	10 (7.0)	
	Unknown	18 (7.4)	3 (7.5)	3 (5.0)	12 (8.4)	
Child-Pugh	A	228 (93.8)	36 (90.0)	54 (90.0)	138 (96.5)	0.117 *
Classification	B7	15 (6.2)	4 (10.0)	6 (10.0)	5 (3.5)	
AFP, ng/mL	Median (range)	10.2 (1.2–38,396.4)	25.3 (1.9–31,466.3)	10.9 (2.2–38,396.4)	9.3 (1.2–16,788.3)	0.062 ^†^
	<10	120 (49.4)	14 (35.0)	30 (50.0)	76 (53.1)	0.127 *
	≥10	123 (50.6)	26 (65.0)	30 (50.0)	67 (46.9)	
Tumor size, cm	Median (range)	2.2 (1.0–17)	6.0 (1.3–17)	3.6 (1.0–12)	1.5 (1.0–12.7)	<0.001 ^†^
	≤2	115 (47.3)	1 (2.5)	16 (26.7)	98 (68.5)	<0.001 *
	>2	128 (52.7)	39 (97.5)	44 (73.3)	45 (31.5)	
TVT	No	184(75.7)	11 (27.5)	40 (66.7)	133 (93.0)	<0.001 *
	Branch	29 (11.9)	7 (17.5)	15 (25.0)	7 (4.9)	
	Main	30 (12.3)	22 (55.0)	5 (8.3)	3 (2.1)	
mUICC stage	I	13 (5.3)	1 (2.5)	2 (3.3)	10 (7.0)	<0.001 *
	II	74 (30.5)	1 (2.5)	13 (21.7)	60 (42.0)	
	III	106 (43.6)	12 (30.0)	31 (51.7)	63 (44.1)	
	IVA	50 (20.6)	26 (65.0)	14 (23.3)	10 (7.0)	
BCLC stage	A	97 (39.9)	0 (0)	17 (28.3)	80 (55.9)	<0.001 *
	B	86 (35.4)	11 (27.5)	22 (36.7)	53 (37.1)	
	C	60 (24.7)	29 (72.5)	21 (35.0)	10 (7.0)	
Diagnosis at PBT	Primary	10 (4.1)	5 (12.5)	2 (3.3)	3 (2.1)	0.021
	Recurrence	233 (95.9)	35 (87.5)	58 (96.7)	140 (95.9)	
Pre-Tx to PBT site	No	52 (21.4)	7 (17.5)	4 (6.7)	41 (28.7)	0.002 *
	Yes	191 (78.6)	33 (82.5)	56 (93.3)	102 (71.3)	
	LRT	186 (97.4)	27 (81.8)	54 (96.4)	102 (97.4)	
	LRT + sorafenib	5 (2.6)	3 (9.1)	2 (3.6)	0 (0)	
	Sorafenib ± chemo	3 (1.6)	3 (9.1)	0 (0)	0 (0)	
Pre-Tx to other site	No	70 (28.8)	22 (55.0)	23 (38.3)	25 (17.5)	<0.001 *
	Yes	173 (43.9)	18 (45.0)	37 (61.7)	118 (82.5)	
	LRT	171 (98.3)	18 (100)	36 (97.3)	117 (98.3)	
	LRT + sorafenib	3 (1.7)	0 (0)	1 (2.7)	2 (1.7)	
Concurrent Tx	No	236 (97.1)	34 (85.0)	59 (98.3)	143 (100)	<0.001 *
	Sorafenib	7 (2.9)	6 (15.0)	1 (1.7)	0 (0)	
Post-Tx to PBT site	No	195 (80.2)	12 (30.0)	48 (80.0)	135 (94.4)	<0.001 *
	Yes	48 (19.8)	28 (70.0)	12 (20.0)	8 (5.6)	
	LRT	16 (33.3)	7 (25.0)	7 (58.3)	2 (25.0)	
	LRT ± sorafenib ± chemo	7 (14.6)	6 (21.4)	0 (0)	1 (12.5)	
	Sorafenib ± chemo	25 (52.1)	15 (53.6)	5 (41.7)	5 (62.5)	
Post-Tx to other site	No	66 (27.2)	8 (20.0)	18 (30.0)	40 (28.0)	0.515 *
	Yes	177 (72.8)	32 (80.0)	42 (70.0)	103 (72.0)	
	LRT	91 (51.7)	4 (12.5)	19 (45.2)	68 (66.0)	
	LRT ± sorafenib ± chemo	57 (32.4)	17 (51.1)	15 (35.7)	26 (25.2)	
	Sorafenib ± chemo	28 (15.9)	11 (34.4)	8 (19.0)	9 (8.7)	

Abbreviations: LC, liver cirrhosis; HBV, hepatitis B virus; HCV, hepatitis C virus; AFP, α-fetoprotein; ECOG, Eastern Cooperative Oncology Group; PS, performance status; TVT, tumor vascular thrombosis; mUICC stage, modified International Union Against Cancer stage; BCLC stage, Barcelona Clinic Liver Cancer stage; Tx, treatment; PBT, proton beam therapy; LRT, locoregional treatment including surgical resection, radiofrequency ablation, percutaneous ethanol injection, transarterial chemoembolization, and radiotherapy; chemo, systemic chemotherapy. * Fisher’s exact test, two-tail. ^†^ One-way analysis of variance.

**Table 2 cancers-11-00230-t002:** Primary tumor and tumor vascular thrombosis (TVT) response according to dose-fractionation regimens and pre- and post-treatment (Tx).

Response		Dose-Fractionation Regimen, *n* (%)			*p* Value *	Pre-Tx to PBT Site, *n* (%)		*p* Value *	Post-Tx to PBT Site, *n* (%)		*p* Value *
		Regimen A	Regimen B	Regimen C		No	Yes		No	Yes	
Primary tumor	CR	16 (40.0)	51 (85.0)	132 (92.3)	<0.001	47 (90.4)	152 (79.6)	0.405	174 (89.2)	25 (53.1)	<0.001
(*n* = 243)	PR	18 (45.0)	6 (10.0)	6 (4.2)		4 (7.7)	26 (13.6)		16 (8.2)	14 (29.2)	
	SD	6 (15.02)	2 (3.3)	5 (3.5)		1 (1.9)	12 (6.3)		5 (2.6)	1 (2.1)	
	PD	0 (0.0)	1 (1.7)	0 (0.0)		0 (0.0)	1 (0.5)		0 (0.0)	1 (2.1)	
TVT	CR	9 (31.0)	15 (75.0)	6 (60.0)	0.021	3 (42.9)	27 (51.9)	0.877	15 (60.0)	15 (44.1)	0.610
(*n* = 59)	PR	12 (41.4)	3 (15.0)	3 (30.0)		3 (42.9)	15 (28.9)		7 (28.0)	11 (32.4)	
	SD	8 (27.6)	1 (5.0)	1 (10.0)		1 (14.2)	9 (17.3)		3 (12.0)	7 (20.6)	
	PD	0 (0.0)	1 (5.0)	0 (0.0)		0 (0.0)	1 (1.9)		0 (0.0)	1 (2.9)	

Abbreviations: CR, complete response; PR, partial response; SD, stable disease; PD, progressive disease; the others are the same as Table 1. * Fisher’s exact test.

**Table 3 cancers-11-00230-t003:** Univariate and multivariate analysis of factors associated with overall survival (OS).

Characteristics		Univariate ^†^					Multivariate ^‡^	
		1-yr OS, % (95% CI)	3-yr OS, % (95% CI)	5-yr OS, % (95% CI)	Median OS, Months (95% CI)	*p* Value	Hazard Ratio (95% CI)	*p* Value
Gender	Male	87.7 (83.2–92.2)	62.2 (55.1–69.3)	48.3 (38.5–58.1)	56.5 (45.5–67.2)	0.964	-	NS
	Female	96.9 (90.8–100)	60.2 (42.6–77.8)	49.3 (29.3–69.3)	46.6 (-)		-	
Age, years	<60	88.0 (81.7–94.3)	63.3 (53.3–73.3)	45.0 (29.1–60.9)	56.5 (35.1–77.9)	0.945	-	NS
	≥60	89.5 (84.4–94.6)	60.8 (52.0–69.6)	49.5 (38.3–60.7)	55.0 (44.5–65.5)		-	
ECOG PS	0	88.6 (84.5–92.7)	62.8 (56.1–69.5)	48.6 (39.4–57.8)	56.5 (44.9–68.0)	0.175	-	NS
	1	100 (-)	25.0 (0–65.0)	- (-)	21.2 (4.1–38.4)		-	
Etiology of LC	HBV	88.8 (84.3–93.3)	62.4 (55.0–69.8)	49.4 (39.2–59.6)	56.5 (-)	0.784	-	NS
	Others	89.1 (80.9–97.3)	60.0 (45.5–74.5)	44.9 (26.1–63.7)	55.0 (32.0–78.0)		-	
Child-Pugh Classification	A	90.4 (86.5–94.3)	65.3 (58.6–72.0)	50.5 (41.1–59.9)	60.3 (-)	<0.001	1.000	0.016
	B7	66.7 (42.8–90.6)	0.91 (0–26.0)	0.91 (0–26.0)	17.1 (1.6–32.6)		2.221 (1.162–4.246)	
AFP, ng/mL	<10	92.5 (87.8–97.2)	74.3 (65.7–82.9)	56.3 (41.6–71.0)	NR	<0.001	1.000	0.008
	≥10	85.4 (79.1–91.7)	49.9 (40.5–59.3)	39.6 (28.6–50.6)	34.3 (24.4–44.2)		1.773 (1.158–2.713)	
Tumor size, cm	≤2	94.8 (90.7–98.9)	79.5 (71.5–87.5)	64.4 (49.3–7.5)	NR	<0.001	-	NS
	>2	83.6 (77.1–90.1)	46.5 (37.1–55.9)	34.0 (23.6–44.4)	33.9 (28.4–39.5)		-	
TVT	No	93.5 (90.0–97.0)	67.8 (60.4–75.3)	54.1 (43.9–64.3)	60.3 (-)	<0.001	–	NS
	Branch	79.3 (60.4–91.4)	49.0 (29.8–68.2)	49.0 (29.8–68.2)	34.3 (-)			
	Main	73.3 (57.4–89.2)	38.8 (21.0–56.6)	18.9 (0–38.1)	19.4 (6.1–32.8)		-	
mUICC stage	I	100 (-)	92.3 (77.8–100)	69.2 (28.6–100)	NR	<0.001	1.000	
	II	94.6 (89.5–99.7)	83.9 (74.5–93.3)	65.4 (45.8–85)	NR		3.186 (0.699–14.525)	0.134
	III	93.4 (88.7–98.1)	57.0 (46.8–67.2)	43.8 (31.3–56.3)	46.6 (34.9–58.4)		6.563 (1.557–27.669)	0.010
	IVA	68.0 (55.1–80.9)	33.4 (19.7–47.1)	26.6 (12.7–40.5)	19.4 (10.0–28.8)		7.119 (1.673–30.288)	0.008
BCLC stage	A	96.9 (93.4–100)	80.1 (71.5–88.7)	65.1 (50.2–80.0)	NR	<0.001	-	NS
	B	89.5 (83.0–96.0)	53.9 (42.3–65.5)	40.0 (26.1–53.9)	37.1 (24.2–50.1)		-	
	C	75.0 (64.0–86.0)	44.6 (31.5–57.7)	32.2 (16.0–48.6)	33.9 (16.2–51.7)		-	
Diagnosis at PBT	Primary	90.0 (71.4–100)	56.0 (22.5–89.5)	42.0 (7.5–76.5)	38.4 (8.5–68.2)	0.578	-	NS
	Recurrence	88.8 (83.8–92.1)	62.0 (55.3–68.7)	48.2 (38.8–57.6)	56.5 (45.3–67.7)		-	
Pre-Tx to PBT site	No	98.1 (94.4–100)	82.3 (71.1–93.5)	78.4 (65.3–91.5)	NR	<0.001	-	NS
	Yes	86.4 (81.5–91.3)	56.8 (49.4–64.2)	41.7 (32.1–51.3)	46.9 (33.7–60.1)		-	
Pre-Tx to other site	No	87.1 (79.3–94.9)	54.0 (41.3–66.7)	41.4 (26.3–56.5)	38.4 (19.7–57.1)	0.130	-	NS
	Yes	89.6 (85.1–94.1)	64.9 (57.3–72.5)	51.2 (39.8–62.5)	NR		-	
Concurrent Tx	No	89.8 (85.9–93.7)	62.8 (56.1–69.5)	49.4 (38.0–60.8)	56.5 (-)	0.001	-	NS
	Sorafenib	57.1 (20.4–93.8)	28.6 (0–62.1)	0 (-)	19.4 (0–40.3)		-	
Post-Tx to PBT site	No	93.8 (90.5–97.1)	67.5 (60.2–74.8)	53.8 (42.6–65.0)	60.3 (-)	<0.001	-	NS
	Yes	68.7 (55.6–81.8)	40.2 (25.9–54.5)	26.2 (12.5–39.9)	25.4 (7.1–43.7)		-	
Post-Tx to other sites	No	81.8 (72.6–91.0)	68.8 (56.8–80.8)	61.9 (45.2–78.6)	NR	0.294	-	NS
	Yes	91.5 (87.3–95.6)	59.8 (52.2–67.4)	44.3 (34.1–54.5)	48.7 (36.3–61.1)		-	
Dose-fractionation	Regimen A	75.0 (61.7–88.3)	33.3 (18.4–48.2)	16.7 (3.6–29.8)	23.4 (15.3–31.5)	<0.001	2.045 (1.244–3.362)	0.005
	Regimen B	86.7 (78.1–95.3)	51.2 (38.1–64.3)	39.2 (24.9–53.5)	40.8 (19.7–61.9)		1.298 (0.740–2.277)	0.363
	Regimen C	93.7 (89.8–97.6)	76.0 (68.4–83.6)	67.9 (57.5–78.3)	NR		1.000	
Primary tumor response	CR	97.0 (94.6–99.4)	73.3 (66.6–80.0)	60.9 (51.5–70.3)	NR	<0.001	1.000	<0.001
	Non-CR	52.3 (37.6–67.0)	10.6 (1.0–20.2)	0 (-)	13.7 (10.8–16.6)		7.012 (4.324–11.370)	

Abbreviations: yr, year; NR, not reached; NS, not significant; CI, confidence interval; the others are the same as in Table 1 and Table 2. ^†^ Log-rank test. ^‡^ Cox proportional hazards model.

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
