# Peer review of "Does Risk-Adapted Proton Beam Therapy Have a Role as a Complementary or Alternative Therapeutic Option for Hepatocellular Carcinoma?"

_cancers, 2019, doi:10.3390/cancers11020230_

Round 1
Reviewer 1 Report
The authors evaluated their original protocols of PBT for HCCs and reported an efficacy of all stages of HCC. The paper provides very interesting clinical data. But from the reviewer point of view, there are many issues in the methods that should be made clear.
My major comments are as follows.
The authors reported good treatment results about regimen B and C.
However, it is not so about regimen A.
HCCs in regimen A are larger than others. This might affect the bad local control. As much as it is a big HCC, it is more likely to be close to GI tract. Furthermore, large HCC is more radioresistant. When there are weak irradiated doses to large HCC, it is natural that the local control rate worsens.
The authors should discuss about the suitable treatment dose for HCC adjacent to the GI tract.
For example, using the following articles:
Nakayama H, et al. Proton beam therapy for hepatocellular carcinoma located adjacent to the alimentary tract. Int. J. Radiat. Oncol. Biol. Phys. 2011;80:992–995.
[19] Mizuhata, M. et al. Respiratory-gated proton beam therapy for hepatocellular carcinoma adjacent to the gastrointestinal tract without fiducial markers. Cancers (Basel) 2018, 10.
Statistical methods are unclear in table 1 and 2. They should be written in the text (page 14; section 4.3). There should not be included in only the table’s abbreviations. It is unknown whether the authors used the comparison between two groups or multiple comparison.
In table 3, Fisher’s exact test should be used the comparison between two categories. The most common use of Fisher’s exact test is for 2x2 tables, so how to use these four caterories (CR, PR, SD and PD)?
In Fig.3, the authors show the statistically differences in Fig.3a, 3c and 3d. However, the log rank statistics fails when the two survival curves cross. Therefore, these results don’t make sense.
Minor comments
Fig 3 each image of graph is unclear.
References
Page 15, line 98
The title of the no.1 Ref is wrong. There is a lack of the character.
This is the [EASL Clinical Practice Guidelines: Management of hepatocellular carcinoma.]
Lines 101-104
In no.2-3, these two guidelines are published from Korea.
The Cancers is an international journal. One of two should quote the guidelines on other countries or global standards.
Author Response
Point-by-point Responses to Reviewers’ comments
Editorial office comment
During our initial check, we noticed that full name of the ethical committee is lacked in your manuscript. Please add it in the Methods Section of the article during the revision.
à Response 1: According to your comments, we revised the manuscript as follow,
In Page 8 line 9, (First paragraph in 4.1 Patients of Materials and Methods)
“… National Cancer Center (NCC) [2];…”
In Page 8 line 16 -17, (First paragraph in 4.1 Patients of Materials and Methods)
“This study was approved by institutional review board of NCC (NCC20180100), and…”
Reviewer #1
The authors evaluated their original protocols of PBT for HCCs and reported an efficacy of all stages of HCC. The paper provides very interesting clinical data. But from the reviewer point of view, there are many issues in the methods that should be made clear.
My major comments are as follows.
The authors reported good treatment results about regimen B and C. However, it is not so about regimen A. HCCs in regimen A are larger than others. This might affect the bad local control. As much as it is a big HCC, it is more likely to be close to GI tract. Furthermore, large HCC is more radioresistant. When there are weak irradiated doses to large HCC, it is natural that the local control rate worsens.
The authors should discuss about the suitable treatment dose for HCC adjacent to the GI tract.
For example, using the following articles:
Nakayama H, et al. Proton beam therapy for hepatocellular carcinoma located adjacent to the alimentary tract. Int. J. Radiat. Oncol. Biol. Phys. 2011;80:992–995.
[19] Mizuhata, M. et al. Respiratory-gated proton beam therapy for hepatocellular carcinoma adjacent to the gastrointestinal tract without fiducial markers. Cancers (Basel) 2018, 10.
à Response 2: We appreciate you for your kind and insightful comments on our paper. According to the reviewers’ suggestions, we added ref 22 (Nakayama H, et al. ) and revised the manuscript as follow,
Page 7 line 11 – 20 (3rd Pargraph in Discussion)
“In other studies[19,21], PBT using relative long fractionation regimens (53.7 – 88 GyE10/24-38 fractions or 80.1 - 91.5 GyE10/22-35 fractions, respectively) showed the 2-year LPFS rate of 94% and 3-year LPFS of 88.1%, respectively, and grade ≥3 GI toxicity of 2.5% and 2.1%, respectively. In present study, there were significance differences in patient characteristics including tumor size, stage, extent of TVT, etc., among three dose-fractionation regimens (Table 1). Thus, large HCC was more likely to close to GI organs and treat with regimen A rather than regimen B and C, and subsequently poor local tumor control in regimen A than regimen B and C. In addition, the incidence of grade ≥3 GI toxicity in the patients treated with regimen A was as low as 2.5%, These findings suggested that it might be possible to carefully escalate the radiation doses for patients with HCC <1 cm from the GI organs to improve tumor control and survival.”
Page 13 line 13 -15, (ref #22)
“22. Nakayama, H.; Sugahara, S.; Fukuda, K.; Abei, M.; Shoda, J.; Sakurai, H.; Tsuboi, K.; Matsuzaki, Y.; Tokuuye, K., Proton beam therapy for hepatocellular carcinoma located adjacent to the alimentary tract. Int J Radiat Oncol Biol Phys 2011, 80, 992-995.”
Statistical methods are unclear in table 1 and 2. They should be written in the text (page 14; section 4.3). There should not be included in only the table’s abbreviations. It is unknown whether the authors used the comparison between two groups or multiple comparison.
In table 2, Fisher’s exact test should be used the comparison between two categories. The most common use of Fisher’s exact test is for 2x2 tables, so how to use these four caterories (CR, PR, SD and PD)?
à Response 3: According to reviewer’s comments, we added the statistical methods in Text (Page 10 line 6 -9). We received the statistical consultation in our institutional statistical office. In manual calculation, Fisher’s exact test can calculate p-value, but, in Statistical Package (STATA, SPSS, SAS, etc.), Fisher’s exact test can calculate the p-value for more than 2x2 tables (such as, 2x3, 3x4 tables, etc.). During statistical consultation, we found the error in Table 2, we revised the manuscript and Table 2 as follow,
Page 10 line 11 – 14, (4.3. Evaluation and statistical considerations in Methods and Materials)
“The distributions of clinical characteristics among dose-fractionation regimens ware compared using Fisher’s exact test or One-way analysis of variance and the differences of primary tumor and TVT responses according to dose-fractionation regimens were assessed using Fisher’s exact test.”
In Table 2,
Pre-Tx to PBT site, n (%) | ||||
Response | No | Yes | P value* | |
Primary tumor | CR | 47 (90.4) | 152 (79.6) | 0.405 |
(n=243) | PR | 4 (7.7) | 26 (13.6) | |
SD | 1 (1.9) | 12 (6.3) | ||
PD | 0 (0.0) | 1 (0.5) | ||
TVT | CR | 3 (42.9) | 27 (51.9) | 0.877 |
(n=59) | PR | 3 (42.9) | 15 (28.9) | |
SD | 1 (14.2) | 9 (17.3) | ||
PD | 0 (0.0) | 1 (1.9) | ||
In Fig.3, the authors show the statistically differences in Fig.3a, 3c and 3d. However, the log rank statistics fails when the two survival curves cross. Therefore, these results don’t make sense.
à Response 4: We appreciate you for your insightful comments on our paper. In general, intersections of survival curves do not always mean not statistical significances. For example, although there were small cross in survival curves in early follow-up period in Fig 3a, the differences of comparing each groups were statistically significant (A vs. B, p=0.033; and B vs. C, p = <0.001). Conversely, there was cross two survival curves in Fig 3c (mUICC stage I and II) and in Fig 3d (BCLC stage B and C), and these survival differences in these subgroups were not statistically significant (>0.05, each). However, except for aforementioned groups, the survival differences between other subgroups (mUICC II vs, mUICC III, mUICC III vs mUICC IVA, BCLC A vs. B, and BCLC A vs. C) and overall survival differences according these factors were statistically significant (<0.05, each). In addition, we intended to show overall statistical significance of survival according to several factors (such as dose-fractionation regimens (A, B, C), tumor response (CR, non-CR), mUICC stage (I, II, III, IVA), and BCLC stage (A, B, C) rather than to compare the each survival curves according to these factors in details.
Minor comments
Fig 3 each image of graph is unclear.
àResponse 5: According to the reviewer’s comment, we revised the Fig 3. Each image of graph is more clear than previous one.
References
Page 15, line 98
The title of the no.1 Ref is wrong. There is a lack of the character.
This is the [EASL Clinical Practice Guidelines: Management of hepatocellular carcinoma.]
àResponse 6: According to your comment, we revised the The title of the no. 1 ref.
Page 11 line 8 - 9, (Ref #1)
“1. European Association for the Study of the Liver.; EASL Clinical Practice Guidelines: Management of hepatocellular carcinoma. J Hepatol 2018, 69, 182-236.”
Lines 101-104
In no.2-3, these two guidelines are published from Korea.
The Cancers is an international journal. One of two should quote the guidelines on other countries or global standards.
à Response 7: Ref #1 is European guideline, and, according to reviewer’s comment, we added the ref #2 (AASLD guideline) as follow,
Page 11 line 10 -12 (ref #2)
“2. Heimbach, J.K.; Kulik, L.M.; Finn, R.S.; Sirlin, C.B.; Abecassis, M.M.; Roberts, L.R.; Zhu, A.X.; Murad, M.H.; Marrero, J.A., Aasld guidelines for the treatment of hepatocellular carcinoma. Hepatology 2018, 67, 358-380.”

Reviewer 2 Report
The paper is presented with knowledge and expertise in the field and reported interesting data regarding thei institutional experience use of proton therapy in hepatocellular carcinoma. I have just a few suggestions/comments: 1) In multivariate analysis primary tumor response was significantly associated with OS. Do you have any data (not reported in the paper) about the time of CR? IF so It would be useful to add them in the manuscript knowing that in HCC patient the CR time can be delayed and it is not standardized. 2) Do you have any data regarding the rate of failure in the low risk volume (regimen A-B)? If so, it would be helpful to add a comment on it in the paper and link it to your proper comment regarding the possibility of escalate the dose for patients with HCCAuthor Response
Point-by-point Responses to Reviewers’ comments
Editorial office comment
During our initial check, we noticed that full name of the ethical committee is lacked in your manuscript. Please add it in the Methods Section of the article during the revision.
à Response 1: According to your comments, we revised the manuscript as follow,
In Page 8 line 9, (First paragraph in 4.1 Patients of Materials and Methods)
“… National Cancer Center (NCC) [2];…”
In Page 8 line 16 -17, (First paragraph in 4.1 Patients of Materials and Methods)
“This study was approved by institutional review board of NCC (NCC20180100), and…”
Reviewer #2
The paper is presented with knowledge and expertise in the field and reported interesting data regarding thei institutional experience use of proton therapy in hepatocellular carcinoma. I have just a few suggestions/comments:
1) In multivariate analysis primary tumor response was significantly associated with OS. Do you have any data (not reported in the paper) about the time of CR? IF so It would be useful to add them in the manuscript knowing that in HCC patient the CR time can be delayed and it is not standardized.
à Response 2: We appreciate you for your kind and insightful comments on our paper. In our previous study regarding with optimal time of CR (ref #13), we showed that most patients (93.9%) showed CR within 12 months and remaining 6.1% patients reached CR up to about 22 months after PBT. In addition, there was no significant differences in LPFS, RFS, OS rates between the patients who reached CR within 6 months after completion of RT and those who reached CR after 6 months. Thus, we evaluated the maximal tumor response observed during the follow-up period (Page 10 line 1 – 3, First paragraph in 4.3. Evaluation and statistical considerations of Materials and Methods). Similar to previous our previous study (ref #13), in present study, median times to CR of the primary tumor and TVT after PBT were 4.5 months (range 1-21.7 months) and 5.1 months (range 1.1-16.4 months), respectively. (In Page 4 line 10 – 12, 2nd paragraph in Results)
2) Do you have any data regarding the rate of failure in the low risk volume (regimen A-B)? If so, it would be helpful to add a comment on it in the paper and link it to your proper comment regarding the possibility of escalate the dose for patients with HCC
àResponse 3: We appreciate you for your kind and insightful comments on our paper. In present study, more advanced and large HCC was more likely to close GI organs and treat with regimen A rather than regimen B and C. These finding might influence the high rate of failure in regimen A rather than regimen B and C. As reviewer’s comment, we discuss this point and revised the manuscript as follow,
Page 7 line 11 – 20 (3rd Pargraph in Discussion)
“In other studies[19,21], PBT using relative long fractionation regimens (53.7 – 88 GyE10/24-38 fractions or 80.1 - 91.5 GyE10/22-35 fractions, respectively) showed the 2-year LPFS rate of 94% and 3-year LPFS of 88.1%, respectively, and grade ≥3 GI toxicity of 2.5% and 2.1%, respectively. In present study, there were significance differences in patient characteristics including tumor size, stage, extent of TVT, etc., among three dose-fractionation regimens (Table 1). Thus, large HCC was more likely to close to GI organs and treat with regimen A rather than regimen B and C, and subsequently poor local tumor control in regimen A than regimen B and C. In addition, the incidence of grade ≥3 GI toxicity in the patients treated with regimen A was as low as 2.5%, These findings suggested that it might be possible to carefully escalate the radiation doses for patients with HCC <1 cm from the GI organs to improve tumor control and survival.”
Page 13 line 13 -15, (ref #22)
“22. Nakayama, H.; Sugahara, S.; Fukuda, K.; Abei, M.; Shoda, J.; Sakurai, H.; Tsuboi, K.; Matsuzaki, Y.; Tokuuye, K., Proton beam therapy for hepatocellular carcinoma located adjacent to the alimentary tract. Int J Radiat Oncol Biol Phys 2011, 80, 992-995.”

Round 2
Reviewer 1 Report
This paper is an important contribution and I recommend that it be accepted for publication.